# The Impact of Sex and Biological Maturation on Physical Fitness in Adolescent Badminton Players

**DOI:** 10.3390/sports11100191

**Published:** 2023-10-03

**Authors:** Jaime Fernandez-Fernandez, Alba Herrero-Molleda, Francisco Álvarez-Dacal, Jose Luis Hernandez-Davó, Urs Granacher

**Affiliations:** 1Department of Physical Education and Sports Sciences, Universidad de León, 24007 León, Spain; jaime.fernandez@unileon.es (J.F.-F.); aherm@unileon.es (A.H.-M.); 2AMRED, Human Movement and Sports Performance Analysis, Universidad de León, 24007 León, Spain; 3Regional Badminton Technification Center (CTD), 33006 Oviedo, Spain; frandacal@badmintonasturias.com; 4Faculty of Health Sciences, Universidad Isabel I de Castilla, 09003 Burgos, Spain; jlhdez43@gmail.com; 5Department of Sport and Sport Science, Exercise and Human Movement Science, University of Freiburg, Sandfangweg 4, 79102 Freiburg, Germany

**Keywords:** racket sports, specific movement, neuromuscular performance, testing

## Abstract

The main objective of this study was to examine the impact of maturity status and sex on selected measures of physical fitness in adolescent badminton players. Eighty-one badminton players (39 boys, 42 girls; age: 12.7 ± 1.4 years; body height: 153.5 ± 10.5 cm; body mass: 48.3 ± 13.2 kg) participated in the study and were divided into pre-peak height velocity (PHV, *n* = 31), circa-PHV (*n* = 29), and post-PHV (*n* = 21) groups. The assessment of physical fitness included linear sprint (5-m, 10-m) and change-of-direction (CoD) speed tests using a modified 5-0-5 CoD test (CoD deficit [CoDD%]) and an on-court CoD test, as well as the countermovement jump (CMJ) test as a proxy of lower limbs’ muscle power. Pre-PHV players presented lower performance levels (*p* < 0.001; ES: 1.81–1.21) than post-PHV in CMJ, linear sprint (5, 10-m) speed, and both CoD tests. In addition, compared to circa-PHV, pre-PHV players demonstrated moderately lower performances in the 10 m sprint and CoD tests (*p* < 0.05; ES: 0.65–1.00). Regarding the CoDD%, no between-group differences were found. Irrespective of the maturity status, boys outperformed girls in CMJ (*p* = 0.01; ES: 0.71), linear sprint speed (*p* < 0.05, ES: 0.52–0.77), and the modified 505 test (*p* = 0.01; ES: 0.71). Findings only showed significant sex-by-maturity interactions for the pre-PHV group. In addition, sex-related performance differences were found in favor of the boys for most measures except for CoDD%. Our results imply that maturity status (i.e., PHV) and not chronological age should be used to design training programs. Female adolescent badminton players should receive specifically targeted exercise interventions based on their fitness status and needs.

## 1. Introduction

The lack of knowledge on sex-related differences (i.e., maturity level, training/competition loads, and injury incidence) may limit the ultimate performance potential of young female athletes. For example, in badminton, coeducational training is often carried out. Coeducation means that female and male adolescent athletes exercise in one training group. A limitation of this approach is that sex-related physiological and maturational differences are seldom considered during exercise programming. Accordingly, training loads are often imbalanced and not individualized [1]. Moreover, coaches often focus on the athletic development of boys and to a lesser extent that of girls [2], which may even increase the imbalance between sexes [3]. Recent research indicates that with the onset of puberty, exercise programming should follow the menstrual cycle, which affords an understanding of the female physiology [4]. Thus, one of the main problems in sports environments is the lack of equity. In other words, the needs, preferences, and interests of girls and boys should be considered equally for performance testing and exercise programming.

The design and implementation of athletes’ training programs should be related to the physiological requirements of the individual and the athletic needs of the sport [5]. As an intermittent sport, badminton is characterized by repetitive short periods of exercise (i.e., 1–9 s) and recovery (i.e., low-intensity activities such as standing and walking for 6–15 s) interspersed with longer breaks in play (i.e., “time outs” of 120 s between games) [6]. Despite the previous attempts to describe the physical and physiological demands of elite badminton players [7,8], information about adolescent players’ characteristics is scarce. In this regard, one of the greatest problems of training at these ages is related to the mismatch between the real tempo (chronological age) and the psycho-biological tempo (growth and maturation rate) [3]. Currently, there are validated and efficient procedures based on anthropometric parameters (“somatic maturation”), which allow for the evaluation of maturity status using equations to estimate peak height velocity (PHV). The assessment of age at PHV allows for the differentiation of pre-, around-, and post-pubertal athletes [9]. Normal maturing girls on average reach PHV at 12 years of age and boys at age 14. Chronological age is therefore not well-suited to allocate boys and girls to one training group. While this is well known in research, many coaches still exercise girls and boys in groups according to their chronological age.

Although the variations in performance could be related to differences in body mass, muscle mass, and overall strength [10], it is unknown whether these differences may affect the physical performance of male versus female badminton players. There is only one previous study [11] with a small sample of 21 badminton players that observed better performances in male versus female players. The results of this study also showed that the sex-related differences increased with increasing age. Furthermore, different researchers [12,13,14] recently showed that in tennis, boys circa-PHV boys aged 15 years showed better performances compared with circa-PHV girls aged 12.5 years in sprint, jump, and change-of-direction (CoD) performances. The observed sex-related differences were more prevalent in players older than 13 years. Given the rising number of badminton competitions from an early age (i.e., under 12 years), knowledge of the fitness profiles of adolescent athletes and the identification of sex differences via the use of appropriate fitness test batteries is a valuable source for talent selection and development.

The aim of this study was to examine the physical fitness (i.e., vertical jump performance, linear sprint, and CoD speed) of adolescent male and female badminton players according to their maturity status. We hypothesized that males and more mature youth would outperform females and less mature youth [12,13,14].

## 2. Methods

### 2.1. Experimental Approach

The current study represents an observational and descriptive analysis to determine performance differences according to the maturity status of adolescent male and female badminton players. Testing protocols were conducted over one week at the end of October 2022, and sessions were undertaken between noon and 5:00 p.m. All players were tested in the facilities of their badminton clubs. All tests were performed in the same order using the same testing devices, measurement protocols, and experienced evaluators. Testing took place on an indoor synthetic court (polyurethane floor; 22.6–23.2 °C; relative humidity, 52–55%; Kestrel 4000 Pocket Weather Tracker, Nielsen Kellerman, Boothwyn, PA, USA). Participants were required to refrain from any intense physical workout for 24 h before the tests and to be in a fasted state for at least 2 h.

### 2.2. Participants

Eighty-one adolescent badminton players were enrolled in this study (39 boys, 42 girls; age: 12.7 ± 1.4 years; body height: 153.5 ± 10.5 cm; body mass: 48.3 ± 13.2 kg; Table 1). Participants comprised talented players selected by the regional badminton federation coaching staff based on technical–tactical skills and competitive performance. All players participated in, on average, 7.8 ± 1.6 weekly hours of combined badminton and conditioning training and had 5.3 ± 1.9 years of badminton training experience. None of the players reported a history of any orthopedic injuries during the previous 12 months. Before taking part in the study, participants and their parents/guardians were fully informed about the study protocol, and both parents/guardians and participants provided their written informed consent. The local institutional ethics committee (Universidad de León; ETICA-ULE-056-2021) approved the procedures in accordance with the latest version of the Declaration of Helsinki.

### 2.3. Procedures

All athletes were familiarized with the study procedures and assessment routines prior to the start of the study. Training volume was estimated by coaches reporting the amount of training time (i.e., combined badminton and conditioning training) during a 6-week period prior to the assessments. Testing was performed on one day (i.e., countermovement jump [CMJ], linear sprints [5 and 10 m], modified 505 CoD tests, and an on-court CoD test). Prior to physical fitness testing, athletes performed a standardized warm-up for 8–10 min, which consisted of a jump rope activation, general dynamic mobility exercises, multi-directional acceleration runs, and jumps of progressive intensity.

#### 2.3.1. Maturity Status

Body height was measured using a fixed stadiometer (±0.1 cm; Holtain Ltd., Crosswell, UK), sitting height using a purpose-built table (±0.1 cm; Holtain Ltd., Crosswell, UK), and body mass using a digital balance (±0.1 kg; ADE Electronic Column Scales, Hamburg, Germany). Pubertal timing was estimated according to the biological maturation of each individual using a predictive equation previously described in the literature [9]. Using chronological age, standing height, sitting height, and body mass, Mirwald et al. [15] developed an easy-to-administer method based on sex-specific regression equations on how to calculate the age of peak linear growth (age at peak height velocity [APHV]) to predict maturity offset (MO). The MO refers to how close (in years) a child is to reaching PHV. Negative values mean that the individual is pre-PHV and positive values indicate that an individual is post-PHV [15].

#### 2.3.2. Countermovement Jump (CMJ) Test

A bilateral CMJ without arm swing was performed using an optoelectrical jump system (Optojump, Microgate, Bolzano, Italy) according to the procedures previously described [16]. Briefly, players performed the jumps starting in an upright standing position with their hands on their hips. Subsequently, athletes flexed their knees using a self-selected depth and then jumped as forcefully and as high as possible. Each player performed three maximal CMJs interspersed with 45 s of passive recovery. The highest jump height was recorded for each athlete and used for further analysis.

#### 2.3.3. Sprint Test

The time during a 10 m linear sprint (with 5 split times) was measured using single beam photocell gates placed 1.0 m above the ground level (Witty System, Microgate, Bolzano, Italy) following the methods previously described [17]. Each sprint was initiated 0.5 m behind the first photocell gate, which then started a digital timer, and players started in a standing position with their preferred foot 0.5 m behind the timing gate. Each player performed three maximal 10 m sprints with at least two minutes of passive recovery in between the trials, and the best performance was recorded.

#### 2.3.4. Modified 505 Change-of-Direction (CoD) Test

The abilities of the athletes to perform a single, rapid 180° CoD over a 5 m distance was measured using a modified version (stationary start) of the 505 CoD test [17]. Players started in an upright standing position with their preferred foot 0.5 m behind the timing gate (Witty System, Microgate, Bolzano, Italy), followed by accelerating forward at maximal effort until reaching a line placed at a 5 m distance. Two trials of pivoting on both left and right feet were completed and the best time was recorded. One minute of rest was allowed between trials [18]. The CoDD% was calculated according to the following formula: CoDD% t = ([modified 505 time − 10 m sprint time]/10 m sprint time) × 100 [19].

#### 2.3.5. On-Court Change-of-Direction (CoD) Test

Players performed an on-court CoD test according to the procedures previously described [20]. In brief, the test was executed on one half of a regular badminton court, using 5 pairs of photocells (Microgate, Bolzano, Italy), which were mounted on supports at a height of 0.5 and 1 m (Figure 1). Players were instructed to run as fast as possible from the central point (marked with an X in Figure 1) towards the pair of photocells placed on the right side of the forecourt (#1 in Figure 1) and return to the center before they consecutively ran to the next pair of photocells (i.e., 2, 3, 4, and 5). Players had to cross the photocells with their waist (e.g., visually checked by the researchers), using sport-specific displacements (i.e., lateral sidestepping, cross-over stepping motions, and/or forward lunges) before returning to the center court. The test was finished when players returned to the central point of the court. Each player performed three trials and the best time was recorded. Two minutes of rest was allowed between trials.

### 2.4. Statistical Analyses

Descriptive statistics (means and standard deviations) were calculated for each of the variables after the normal distribution was tested and confirmed using the Kolmogorov–Smirnov test. Within-session reliability of test measures were assessed using intraclass correlation coefficients (ICC) and the coefficient of variation (CV) with their corresponding 95% confidence intervals, respectively. We considered an ICC < 0.50 as poor, 0.50 ≤ ICC < 0.75 as moderate, 0.75 ≤ ICC < 0.90 as good, and ICC > 0.90 as excellent [21]. Absolute reliability was calculated using the standard error of measurements (SEM), which was calculated as SD × √1 − ICC, where SD is the SD of all scores from the subjects [22]. The SEM was used for calculating the minimal detectable change (MDC) and was calculated as SEM × 1.96 × √2 to construct a 95% CI [22]. In order to investigate the differences caused by the maturity status, as well as in-between maturity groups (i.e., pre-PHV girls vs. pre-PHV boys; circa-PHV girls vs. circa-PHV boys, and post-PHV girls vs. post-PHV boys), one-way independent measures analyses of variance (ANOVAs) were performed, and the Bonferroni post hoc test was used to aid in the interpretation of the results. Differences between males and females according to maturity status were compared using an independent samples *t*-test. Effect sizes (ES; Cohen’s d) were calculated to estimate the magnitude of differences in the tested variables and interpreted using the following thresholds: <0.2, trivial; ≥0.2 to 0.49, small; ≥0.5 to 0.79, moderate; and ≥0.8, large [23]. Precise *p*-values were reported, and the significance level was set at *p* (probability of type I error) < α = 0.05. All statistical analyses were performed using IBM SPSS Statistics 26.0 (SPSS, Inc., Chicago, IL, USA).

## 3. Results

No test-related injuries were recorded during the experimental period of this study. Table 2 contains test–retest reliability data for the assessed variables. All tests showed acceptable between-trial reliability scores, with CV values ranging between 1.4–3.5% and good to excellent ICCs (0.849 to 0.990) (Table 2).

Pre-PHV players were shorter in body stature (*p* < 0.01; ES: 0.77 and 2.70) and had less body mass (*p* < 0.01; ES: 0.94 and 1.84), together with younger MO (*p* < 0.001; ES: 2.8 and 4.7) compared to circa-PHV and post-PHV players. No statistically significant differences were found between the maturity groups with regard to APHV. Moreover, the circa-PHV group had less body mass (*p* = 0.01; ES: 0.78), lower height (*p* < 0.001; ES: 1.60), and younger MO (*p <* 0.001; ES: 2.79) than the post-PHV group. Regarding training volume, results showed that pre-PHV players presented lower exercise volume compared to circa- (*p* < 0.05; ES: 0.70) and post-PHV players (*p* < 0.001; ES: 2.73). Sex-specific results indicated that boys showed higher chronological ages (< 0.001; ES: 0.88), while girls achieved the APHV earlier (*p* < 0.001; ES: 5.6), Irrespective of the maturity status, boys outperformed girls in CMJ (*p* = 0.01; ES: 0.71) performance, linear sprint speed (5-m: *p* = 0.005, ES: 0.77; 10-m: *p* = 0.05, ES:0.52), and the modified 505 test (*p* = 0.01; ES:0.71). No between-group differences were found for training volume (*p* = 0.17; ES: 0.37), CoDD% (*p* = 0.21; ES: 0.32), and the on-court CoD test (*p* = 0.21; ES: 0.34).

Table 3 shows the differences between maturity groups in vertical jump performance, linear sprint, and CoD speed. The between-group analyses revealed significant differences (*p* < 0.001 to 0.03) for all tests performed (i.e., CMJ, linear sprinting, and CoDs). Post hoc analyses showed that pre-, compared to post-PHV athletes were significantly slower (*p* < 0.001; ES: 1.81 to 1.21) in linear sprint (5 and 10-m) and CoD performances (M505 and on-court test). Moreover, they showed significantly lower performance levels in CMJ height (*p* < 0.001; ES: 1.68). In addition, compared to circa-PHV, pre-PHV players also demonstrated moderately lower performances in 10 m sprint (*p* < 0.05; ES: 0.65) and the M505 test (*p* < 0.05; ES: 0.77), with impaired performance in the on-court CoD test (*p* < 0.05; ES: 1.00). Regarding the CoDD%, no differences were found between groups.

When comparing circa-PHV and post-PHV players, the results showed significantly lower CMJ values (*p* < 0.05; ES: 0.99) in the circa-PHV group. Large between-group differences were found for the 5 m sprint (*p* < 0.05; ES: 1.45) and the on-court CoD test (*p* < 0.01; ES: 0.94) in favor of the more mature group. Post-PHV players showed better CoDD% (*p* < 0.05; ES: 0.94) compared with circa-PHV. No differences between circa- and post-PHV were found for the 10 m sprint and the M505 test.

Table 4 shows differences between maturity groups (i.e., pre-PHV girls vs. pre-PHV boys; circa-PHV girls vs. circa-PHV boys, and post-PHV girls vs. post-PHV boys) according to sex. Although no significant sex-by-maturity interactions were detected, between-group analyses revealed several significant differences. In the pre-PHV group, training volume was higher in boys than girls (*p* < 0.001; ES: 1.91), and larger sex-related differences were reported for all tests (*p* < 0.001; ES: 1.68 to 1.97) except for CoDD%.

## 4. Discussion

The aim of this study was to examine the physical fitness (i.e., vertical jump performance, linear sprint, and CoD speed) of adolescent male and female badminton players according to their maturity status. Our main results indicate that boys presented better CMJ and linear sprint (5–10 m) and CoD performances (M505 test) than girls. No significant sex-related differences were found for training volume and the on-court CoD test. Pre-PHV compared with the post-PHV players presented lower jump (CMJ), linear sprint (5–10 m), and CoD performances (both modified 5-0-5 and the on-court test). In addition, compared with circa-PHV, pre-PHV players also demonstrated lower performance levels in both, 10 m linear sprint and CoD speed. Regarding the CoDD%, no statistically significant differences were found between the maturity groups. Finally, when analyzing sex-by-maturity interactions, results only showed significant interactions for the pre-PHV group, with a higher training volume in boys than girls. In addition, sex-related performance differences were large for most measures, except for CoDD%.

Regarding the observed sex-related differences, our results can be interpreted as an enhanced neuromuscular performance of boys compared with girls. It is likely that this difference is primarily caused by the higher strength capacity of boys (rather than a sex-mediated effect) [24]. It seems that when subjects are strength-matched, absolute sex-related performance differences disappear, as recently reported [10]. Thus, these variations can be caused by distinct strength values (rather than a biological consequence induced by sex). In other words, girls could show similar performance levels as boys if the strength capacity were similar between the sexes. This is especially important since girls are ahead of boys in terms of biological maturation, with present results showing that girls reached the APHV significantly earlier (~years) than boys, suggesting that girls should start earlier with strength and conditioning programs [2]. Future research could test this hypothesis to determine whether the decreased neuromuscular performance (i.e., jump performance, linear sprint, and CoD speed) in girls is a direct result of lower strength levels.

During the early stages of long-term athletic development, players spend a great amount of training time mastering their individual badminton-specific skills, with technical and tactical sessions constituting an important part of the training week. In this regard, the information about training volume in adolescent badminton players is scarce. Only a few studies are available and researchers reported weekly training volumes ranging between 16–20 h hours at ages below 13 years old, with training volume increases up to 25 h per week in competitive players aged 13–15 years [25,26]. Our results showed that the analyzed sample is far from this training volume, which highlights the recreational character of our study sample. Furthermore, when analyzing training volumes, there were no differences between boys and girls, although boys showed older chronological ages (i.e., 13.3 years vs. 12.2 years for boys and girls, respectively), which can be related to a slightly higher experience in specific training. In this regard, boys presented better performance levels in several fitness tests (i.e., CMJ, linear sprint, and CoD performances), which could be related to a bias directed towards a sport-specific activity at the expense of fitness training. Thus, the inclusion of training programs that incorporate a variety of essential motor skills (i.e., locomotion, stabilization, and strength) seems a worthwhile strategy to maximize motor skill proficiency and reduce the risk of sustaining acute and overuse injuries in adolescent athletes [27]. In addition, daily training practices of adolescent badminton players are often characterized by coeducation with boys and girls grouped according to chronological age. In this regard, it is well known that large interindividual biological differences can exist within sexes and in individuals of the same chronological age [3], suggesting that chronological age is not a good indicator to use for training programming. Thus, a practical approach to design individualized training programs, and to organize training groups that are related to certain periods of trainability during the process of maturation is the use of an athlete’s PHV [28]. The rationale for PHV usage is that maturity has a large impact on physical fitness qualities such as linear sprint and CoD speed, and jump performance. For example, we were able to show significant differences in body height and mass according to the maturity status. Pre-PHV players were significantly shorter (>12%) and had less body mass (>20%) than circa-PHV and post-PHV players. Previous researchers reported that biological maturity leads to significant increases in body height and mass [3], which, in some cases, may disrupt motor coordination in complex motor coordination tasks (i.e., “adolescent awkwardness”) [29]. From a practical point of view, although these alterations are not an absolute expectation, they may temporarily compromise the regulation of the lower extremity joint stiffness and lead to impairments in the individual’s ability to control multi-joint movements [30].

Regarding the linear sprint and jumping ability, the use of linear sprint tests or the CMJ as valid tests for the general assessment of lower limbs’ neuromuscular function are well accepted in the literature [31] and seem to be relevant to describe its development during different stages of maturation [32]. For linear sprint speed, pre-PHV players were significantly slower than post-PHV (~10–12%) players, and although non-significant, there was a tendency to be slower than their circa-PHV (~5%) peers. Moreover, circa-PHV players were also significantly slower than their post-PHV counterparts (~8%) in the 5 m sprint, and although non-significant, ~6% slower in the 10 m. To the best of our knowledge, there is no previous study available that analyzed these physical qualities in a similar youth badminton cohort. Therefore, between-study comparisons are not possible. However, our results are in agreement with previous research conducted with adolescent athletes from other sports [14,33,34], showing that pre-PHV players presented lower levels (~10–20%) of jump and sprint performance than their post-PHV peers. The period around PHV seems to be a key point in which the ability to develop vertical power and speed accelerate [29,35], which continue to increase, although not significantly, until the post-PHV stage. Several physiological factors such as increases in muscle size, limb length, or tendon stiffness could be related to the observed between-group differences [36,37,38], although we did not directly measure them in this study.

In badminton, players must frequently change direction towards the center of the court and then towards the opponents’ return [39], highlighting the crucial importance of CoD for this sport. Accordingly, previous researchers reported a significant relationship between CoD speed and the winning percentage in national-level players [40], highlighting this physical quality as one of the most important athletic skills needed to be a successful badminton player at any level. Consequently, the assessment of CoD should be included within badminton players’ test batteries. In the present study, we included a simple CoD test plus a badminton-specific test, which was previously shown to be highly sensitive to discriminate between athletes from different age groups and expertise levels [20]. Our results showed that boys outperformed girls in the M505 test, but there were no differences in the CoDD%. Thus, faster players with linear sprint performance did not present higher CoD deficits, which is in line with a recent study in young tennis players [17]. However, these results are also contrary to previous studies in team-sport athletes (both professional and youth players), showing that faster and more powerful subjects tend to present higher CoD deficits (when compared with their slower and weaker peers) [41,42]. Since badminton players are required to make a great number of directional changes during rallies, with a very short distance covered on each stroke, and rarely reach maximum speed, we can suggest that this group of players tends to be more efficient in this specific task (e.g., modified 505 test). Since the distance covered in a 505 test does seem not enough to achieve considerable velocity, badminton players analyzed here seem to have the ability to handle and tolerate the higher approach velocities throughout this measurement.

Results regarding the on-court CoD test showed that comparing maturation groups, pre-PHV presented lower levels of performance (~15–20%) compared to their circa- and post-PHV peers. These results are in line with a previous study reporting that this test showed differences of large magnitude between U17 and U19 players [20], and reinforcing the idea that the test is highly sensitive to discriminate between players of different levels. Interestingly, when comparing genders, results of the on-court CoD test showed no differences between boys and girls, although performance levels were trivial in girls (~6%; ES: 0.34) compared to the boys. Although girls showed lower performance levels overall, there were more numbers in the circa- and post-PHV stages. Thus, we could hypothesize that together with similar training volume between genders, girls seem to be more efficient in a sport-specific displacement. However, we need more research to confirm this hypothesis.

This study has important implications for daily exercise practice and future research. Considering that daily training practices of adolescent badminton players usually involve coeducation (i.e., girls and boys), the use of maturity status (i.e., PHV) instead of chronological age could be more appropriate when designing training programs. This strategy may be especially useful when players are training and competing in chronological age-based environments in which they may be more susceptible to maturation bias, particularly immature individuals [3]. Moreover, the performance of boys and girls is different, especially in the first periods of sports development (i.e., pre-PHV). Therefore, coaches should not only be aware of these differences but also provide efficient training stimuli to avoid possible fitness deficits. From a practical perspective, it seems that girls benefit the most from strength and conditioning programs incorporated into training before the onset of puberty [43], aiming to improve performance via enhanced neuromuscular activation and also helping to minimize the risk of sustaining injuries.

The findings of this study suggest the need to include specific training strategies related to the different maturation stages. In this regard, badminton training centers or schools can take advantage of using the online available PHV calculation equations (i.e., https://wwwapps.usask.ca/kin-growthutility/phv_ui.php) every 2–3 weeks to design optimal training groups and plan the consequent training loads. Thus, players around PHV can benefit from neuromuscular training programs (i.e., foundational strength and movement skills) [44], which could be beneficial for developing their neuromuscular qualities [45]. However, with the onset of puberty, circulating anabolic hormones (i.e., testosterone) provide an adequate foundation to build muscle mass via heavy resistance and strength exercises [46].

A number of study limitations are worth mentioning, such as its cross-sectional design and the inclusion of bigger samples, including pre-, around, and post-PHV players organized by their competitive levels (regional, national) that would help to avoid potential selection bias. Furthermore, a more detailed analysis of the training volume at different maturation stages including specific volume for strength, endurance, and/or other qualities would help to clarify if performance differences are also mediated by training. The examined research question could also be influenced by external factors (exercise infrastructure) or sociodemographic factors. Future studies could therefore examine the research question in cohorts of different sociodemographic or cultural backgrounds. Finally, the lack of strength/power-related measurements could help to determine whether the differences found herein are mediated by differences in the strength levels.

## 5. Conclusions

To the best of our knowledge, this is the first study that compared physical fitness (i.e., vertical jump performance, linear sprint, and CoD speed) according to the maturity status (i.e., PHV) of adolescent male and female badminton players. We observed that although training volume was similar between groups, boys outperformed girls in CMJ height, linear sprint (5–10 m), and CoD speed (M505 test), with no differences in the on-court CoD test. When comparing players’ maturity status, our results showed that pre-PHV players presented lower levels of performance in jumping ability (CMJ), linear sprint (5–10 m), and CoD speed (both modified 5-0-5 and the on-court test) compared with the post-PHV players. In addition, compared with circa-PHV players, pre-PHV players also demonstrated lower performance levels in both 10 m sprint and CoD performance. Regarding the CoDD%, no significant differences were found between groups of different maturity statuses. Our results imply that biological age (i.e., maturity status) and not chronological age should be used to design training programs. Female adolescent badminton players should receive specifically targeted exercise interventions based on their fitness status and needs.

## Figures and Tables

**Figure 1 sports-11-00191-f001:**
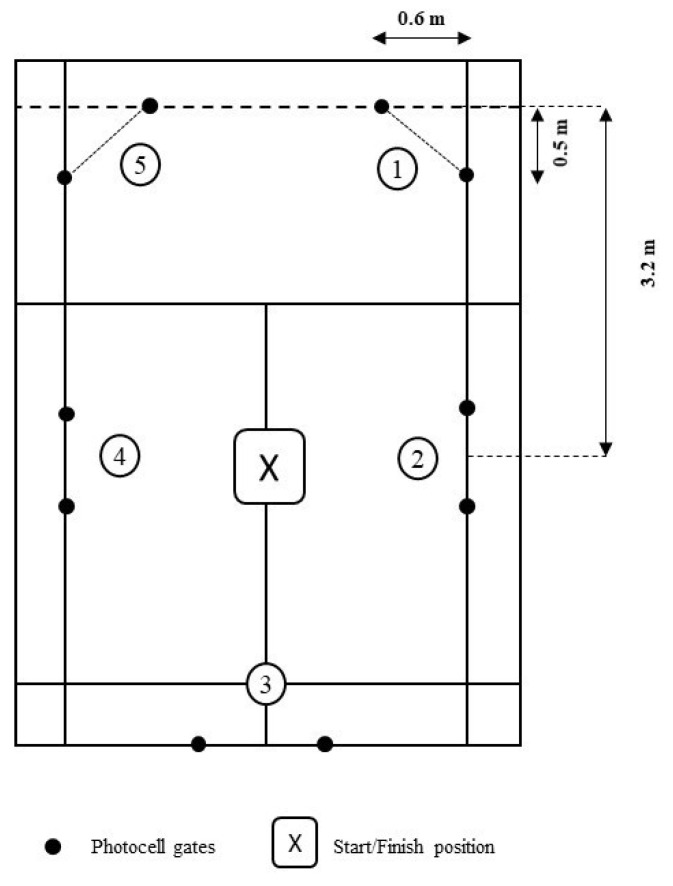
Schematic representation of the on-court change-of-direction (CoD) test.

**Table 1 sports-11-00191-t001:** Descriptive characteristics of the participating badminton players according to their maturity status.

	All Players (*n* = 81)	Girls (*n* = 42)	Boys (*n* = 39)	Maturity Status		
				Pre-PHV (*n* = 31)	Circa-PHV (*n* = 29)	Post-PHV (*n* = 21)
Chronological age (years)	12.7 ± 1.4	12.2 ± 1.3	13.3 ± 1.2 **	11.5 ± 1.3	12.1 ± 0.7	14.0 ± 0.6
Body height (cm)	153.5 ± 10.4	150.6 ± 10.3	157 ± 10.0 *	146.1 ± 8.7 ^$###^	152.9 ± 9.1 ^###^	163.8 ± 3.4
Body mass (kg)	48.3 ± 13.2	41.6 ± 7.2	56.5 ± 13.9 **	39.7 ± 6.8 ^$###^	48.9 ± 12.0 ^##^	58.7 ± 12.9
APHV (years)	13.0 ± 1.0	12.3 ± 0.4	13.9 ± 0.5 **	13.2 ± 1.1	13.0 ± 0.9	12.8 ± 0.7
Maturity offset (years) ^&^	0.4 ± 1.8	−0.1 ± 1.2	−0.6 ± 1.4	−1.9 ± 0.7 ^$$###^	−0.1 ± 0.6 ^###^	2.2 ± 1.0
Training volume (h·week^−1^)	8.0 ± 1.2	7.6 ± 1.8	10.3 ± 4.3	6.8 ± 1.0 ^###^	7.6 ± 1.5 ^###^	9.2 ± 1.2

The values presented are means ± SD. Pre-PHV: Pre-peak height velocity group; Circa-PHV: Around-peak height velocity group; Post-PHV: Post-peak height velocity group; APHV: Estimated age at peak height velocity. ^&^ Estimation of years from predicted PHV. * Significantly different from females (*p* < 0.05); ** Significantly different from females (*p* < 0.001). ^$^ Significantly different from circa-PHV group (*p* < 0.05). ^$$^ Significantly different from circa-PHV group (*p* < 0.001). ^##^ Significantly different from post-PHV group (*p* < 0.01). ^###^ Significantly different from post-PHV group (*p* < 0.001).

**Table 2 sports-11-00191-t002:** Within-session reliability of the applied test measurements.

	ICC (95% CI)	CV (%) (95% CI)	SEM	MDC
**Vertical jumping ability**				
CMJ (cm)	0.985 (0.974–0.992)	3.45 (2.65–4.24)	1.12	3.10
**Linear sprint speed**				
5 m (s)	0.975 (0.948–0.988)	2.34 (1.70–2.98)	0.03	0.09
10 m (s)	0.990 (0.981–0.995)	1.36 (0.88–1.95)	0.05	0.13
**Change-of-direction speed**				
M505 (s)	0.925 (0.837–0.961)	2.23 (1.77–2.70)	0.10	0.28
CoDD%	0.957 (0.871–0.986)	2.44 (1.62–3.25)	0.05	0.13
On-Court test (s)	0.849 (0.775–0.898)	2.10 (1.58–2.63)	0.21	0.57

ICC = intraclass correlation coefficient; CV = coefficient of variation; CI = confident interval; SEM = standard error of measurement; MDC = minimal detectable change; CMJ: countermovement jump; M505: modified change-of-direction test; CoDD%: Percentage-based CoD deficit.

**Table 3 sports-11-00191-t003:** Between-group differences in measures of physical fitness according to the maturity status.

	Maturity Status			One-Way ANOVA		Effect Size (90% IC)	
	Pre-PHV (*n* = 31)	Circa-PHV (*n* = 29)	Post-PHV (*n* = 21)	*p*-Value	Pre-PHV vs. Circa-PHV	Circa-PHV vs. Post-PHV	Pre-PHV vs. Post-PHV
**Vertical jumping ability**							
CMJ (cm)	22.9 ± 5.7 ^$$$^	26.1 ± 5.7 ^$^	30.6 ± 3.1	<0.001	0.56 (0.05–1.07)	0.99 (0.43–1.56)	1.68 (1.07–2.29)
**Linear sprint speed**							
SP5 (s)	1.23 ± 0.11 ^$$$^	1.17 ± 0.10 ^$$^	1.07 ± 0.10	<0.001	−0.62 (−1.13–−0.11)	−1.45 (−2.06–−0.85)	−1.81 (−2.44–−1.19)
SP10 (s)	2.18 ± 0.21 ^†$$$^	2.06 ± 0.15	1.94 ± 0.12	<0.001	−0.65 (−1.17–0.14)	−1.10 (−1.67–−0.53)	−1.55 (−2.15–−0.94)
**Change-of-direction speed**							
M505 (s)	3.26 ± 0.34 ^†$$$^	3.05 ± 0.20	2.97 ± 0.05	0.01	−0.77 (−1.29–−0.26)	−0.54 (−1.09–−0.00)	−1.21 (−1.79–−0.64)
CoDD%	49.4 ± 6.5 ^†$$$^	47.8 ± 6.0 ^$^	52.8 ± 4.5	0.03	−0.26 (−0.76–0.24)	0.94 (0.38–1.50)	0.60 (0.06–1.15)
On-Court Test (s)	14.71 ± 1.70 ^†$$$^	13.08 ± 1.58 ^$$^	11.13 ± 2.46	<0.001	−1.00 (−1.54–−0.47)	−0.94 (−1.51–−0.38)	−1.70 (−2.32–−1.09)

The values presented are means ± SD. F- and *p*-values were obtained by One-Way ANOVA (3 maturational groups). Pre-PHV: Pre-peak height velocity group; Circa-PHV: Around-peak height velocity group; Post-PHV: Post-peak height velocity group; 90% IC: 90% interval confidence; CMJ: countermovement jump; SP5: 5 m linear sprint; SP10: 10 m linear sprint; M505: modified change-of-direction test.; CoDD%: Percentage-based change-of-direction (CoD) deficit. ^†^ Different from Circa-PHV group (*p* < 0.05). ^$^ Different from Post-PHV group (*p* < 0.05). ^$$^ Different from Post-PHV group (*p* < 0.01). ^$$$^ Different from Post-PHV group (*p* < 0.001).

**Table 4 sports-11-00191-t004:** Between-group differences in measures of physical fitness according to sex.

	Boys			Girls			*p*-Value Effect Size (90% IC)		
	Pre-PHV (*n* = 18)	Circa-PHV (*n* = 11)	Post-PHV (*n* = 7)	Pre-PHV (*n* = 13)	Circa-PHV (*n* = 18)	Post-PHV (*n* = 11)	Pre-PHV	Circa-PHV	Post-PHV
**Vertical jumping ability**									
CMJ (cm)	26.2 ± 4.7	28.1 ± 1.7	31.7 ± 1.6	18.9 ± 4.1	25.0 ± 6.7	29.7 ± 3.9	0.001 −1.68 (−2.50–−0.87)	0.250 −0.63 (−1.41–0.14)	0.217 −0.64 (−1.46–0.18)
**Linear sprint speed**									
SP5 (s)	1.15 ± 0.83	1.16 ± 0.06	1.03 ± 0.02	1.30 ± 0.07	1.17 ± 0.09	1.11 ± 0.04	<0.001 1.83 (0.99–2.66)	0.659 0.22 (−0.54–0.98)	<0.001 1.04 (0.19–1.89)
SP10 (s)	2.05 ± 0.14	2.07 ± 0.10	1.95 ± 0.03	2.34 ± 0.16	2.06 ± 0.17	1.93 ± 0.06	<0.001 1.96 (1.11–2.82)	0.951 −0.03 (−0.79–0.73)	0.468 −0.37 (−1.17–0.43)
**Change-of-direction speed**									
M505 (s)	3.05 ± 0.24	2.99 ± 0.10	2.95 ± 0.04	3.51 ± 0.28	3.07 ± 0.24	2.98 ± 0.03	<0.001 1.81 (0.98–2.65)	0.399 0.46 (−0.31–1.22)	0.159 0.72 (−0.10–1.54)
CoDD%	49.2 ± 7.7	45.0 ± 4.8	51.2 ± 1.2	49.6 ± 5.2	49.2 ± 6.2	54.2 ± 5.8	0.901 0.05 (−0.65–0.76)	0.142 0.74 (−0.04–1.52)	0.169 0.72 (−0.10–1.55)
On-Court Test (s)	13.66 ± 1.50	12.63 ± 0.65	11.32 ± 0.37	15.99 ± 0.74	13.30 ± 1.87	10.96 ± 3.44	<0.001 1.97 (1.11–2.82)	0.369 0.48 (−0.29–1.25)	0.776 −0.15 (−0.94–0.65)

The values presented are means ± SD. F- and *p*-values were obtained by One-Way ANOVA (3 maturational groups). Pre-PHV: Pre-peak height velocity group; Circa-PHV: Around-peak height velocity group; Post-PHV: Post-peak height velocity group; 90% IC: 90% interval confidence; CMJ: countermovement jump; SP5: 5 m linear sprint; SP10: 10 m linear sprint; M505: modified change-of-direction test.; CoDD%: Percentage-based change-of-direction (CoD) deficit.

## Data Availability

The datasets used and/or analyzed during the current study are available from the corresponding author upon reasonable request.

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
