# Peer review of "The Impact of Sex and Biological Maturation on Physical Fitness in Adolescent Badminton Players"

_sports, 2023, doi:10.3390/sports11100191_

Round 1

Reviewer 1 Report

The study title should be "child badminton players" instead of "young badminton players". 12-13 year olds are called children. Maybe you can say, adolescent. To be young, subjects must be over 15 years of age (16-20 years). The calendar ages and training volumes of boys and girls calendar ages and training volumes are different. This difference needs to be taken into account in the interpretations. It should be clearer how the maturity status of children is determined. For people in foreign countries, the current explanation may not be sufficient. Especially girls can be in the puberty stage. For this reason, children could be asked with a questionnaire whether they are in the puberty stage or not. Or if there is no puberty, it should be specified in this method.

Author Response

Reviewer 1:

The study title should be "child badminton players" instead of "young badminton players". 12-13 year olds are called children. Maybe you can say, adolescent. To be young, subjects must be over 15 years of age (16-20 years). The calendar ages and training volumes of boys and girls calendar ages and training volumes are different. This difference needs to be taken into account in the interpretations. It should be clearer how the maturity status of children is determined. For people in foreign countries, the current explanation may not be sufficient. Especially girls can be in the puberty stage. For this reason, children could be asked with a questionnaire whether they are in the puberty stage or not. Or if there is no puberty, it should be specified in this method.

Suggested response:

Thank you for your overall positive rating of our manuscript and your helpful comments. We have addressed the raised issue in a point by point response and highlighted the changes in YELLOW in the revised manuscript.

Comment 1: The study title should be "child badminton players" instead of "young badminton players". 12-13 year olds are called children. Maybe you can say, adolescent. To be young, subjects must be over 15 years of age (16-20 years).

Response to comment 1: The World Health Organization (WHO) defines the age of adolescence as between 10 and 18 years. According to the seminal work of Malina, Bouchard and Bar-Or (Growth, Maturation, and Physical Activity), the age ranges 8-19 years in girls and 10 to 22 years in boys  as limits for the onset and termination of adolescence.

Accordingly, the title was changed to

The impact of sex and biological maturation on physical fitness in adolescent badminton players

Comment 2:

The calendar ages and training volumes of boys and girls calendar ages and training volumes are different. This difference needs to be taken into account in the interpretations.

Response to comment 2:

Comment acknowledged. Although it is true that both, chronological ages, and APHV are significanlty different, in terms of training volume, results showed no significant differences between sexes. We will mention that in the results section, and discussion.  

Comment 3:

It should be clearer how the maturity status of children is determined. For people in foreign countries, the current explanation may not be sufficient. Especially girls can be in the puberty stage. For this reason, children could be asked with a questionnaire whether they are in the puberty stage or not. Or if there is no puberty, it should be specified in this method.

Response to comment 3:

The reviewer is correct which is why we included more information on the Mirwald study (published in MSSE in 2002) in the methods section and how the maturation status was assessed based on anthropometric data (sitting and standing height, body mass).

We included the following information in the methods section:

“Using chronological age, standing height, sitting height, and body mass, Mirwald et al. developed an easy-to-administer method based on sex-specific regression equations on how to predict maturity offset. The maturity offset refers to how close (in years) a child is to reach peak height velocity (PHV). Negative values mean that the individual is pre-PHV, positive values indicate that an individual is post-PHV.”

Reviewer 2 Report

The main aim of this study was to examine the impact of maturity-status and sex on selected measures of physical fitness in young badminton players. Regarding the authors, I would like to congratulate and thank them for their effort and motivation involved in this research study. The presentation of the research is well documented, with a scientific basis and respects the latest standards regarding the highest level scientific publications. The methodology was chosen correctly. The conclusions support and result from the research and open new directions for future research. The submitted work is interesting and essentially exhausts the subject under discussion. I have only a few minor suggestions:

Firstly, for titles in MDPI journals, the main words are capitalized and most secondary words are lowercase (title case: APA style). The title in this manuscript, therefore, needs to be made compliant.

Secondly, the article states that the study was conducted in accordance with the Declaration of Helsinki and World Medical Association, what is extremely important. However, there is no information whether the participants were treated ethically according to the unified American Psychological Association code of ethics? Please complete this information in the manuscript. This is an important element from an ethical point of view.

Thirdly, I find the cultural and socio-political context missing from the manuscript, which can be extremely important in terms of the covered subject matter. Depending on where young people train (in which cultural settings), and what access they have to infrastructure, their physical fitness may be higher or lower. This study is a kind of case study, so to speak, and it would have been useful to deepen the non-sporting aspects as well, for example in the introduction or discussion section.

In addition, please make your conclusions more generalised so that they can also be understood by a reader with no specialist knowledge of sport or physical culture.

Supplementing the article with the above-mentioned scope will in my opinion make a real chance for publication in Sports. I keep my fingers crossed for the final success of the publication. 

Author Response

Suggested responses

Thank you for your overall positive rating of our manuscript and your helpful comments. We have addressed the raised issue in a point by point response and highlighted the changes in GREEN in the revised manuscript.

Comment 1:

The main aim of this study was to examine the impact of maturity-status and sex on selected measures of physical fitness in young badminton players. Regarding the authors, I would like to congratulate and thank them for their effort and motivation involved in this research study. The presentation of the research is well documented, with a scientific basis and respects the latest standards regarding the highest level scientific publications. The methodology was chosen correctly. The conclusions support and result from the research and open new directions for future research. The submitted work is interesting and essentially exhausts the subject under discussion. I have only a few minor suggestions.

Response to comment 1: Thank you very much for your affirmative comment on our manuscript. Very much appreciated.

Comment 2: Firstly, for titles in MDPI journals, the main words are capitalized and most secondary words are lowercase (title case: APA style). The title in this manuscript, therefore, needs to be made compliant.

Response to comment 2: Thanks for informing us. We adapted the title in accordance with MDPI guidelines.

Comment 2: Secondly, the article states that the study was conducted in accordance with the Declaration of Helsinki and World Medical Association, what is extremely important. However, there is no information whether the participants were treated ethically according to the unified American Psychological Association code of ethics? Please complete this information in the manuscript. This is an important element from an ethical point of view.

Response to comment 2: Thank you for the suggestion. Please note that we are exercise physiologists and not psychologists. We double checked the guidelines of the unified American Psychological Association. On the main page of the website, the association states

“Ethical Principles of Psychologists and Code of Conduct code of ethics”

We are not psychologists but we conform to these principles as well. Nevertheless, given that we did not have the psychological ethics principles in mind when the study was designed and conducted, we do not want to include a statement a posteriori that we followed these instructions – even though we coincidentally did. Again, we definitely followed the latest version of the Declaration of Helsinki and we had local ethics approval (Universidad de León; ETICA-ULE-056-2021). We hope the reviewer agrees that these statements are sufficient considering the described circumstances. Thank you.

Comment 3: Thirdly, I find the cultural and socio-political context missing from the manuscript, which can be extremely important in terms of the covered subject matter. Depending on where young people train (in which cultural settings), and what access they have to infrastructure, their physical fitness may be higher or lower. This study is a kind of case study, so to speak, and it would have been useful to deepen the non-sporting aspects as well, for example in the introduction or discussion section.

Response to comment 3: Yes, the reviewer is correct. The sociodemographic context could influence the outcomes of the study. This would be a separate research question for instance through the comparison of this Caucasian study cohort with for example an Asian study cohort to find out whether there are differences in how the different study cohorts adapt to training. This was not the goal of our study which is why we decided to address your point in the study limitations. An additional statement was included:

The examined research question could also be influenced by external factors (exercise infrastructure) or sociodemographic factors. Future studies could therefore examine the research question in cohorts of different sociodemographic or cultural background.

Comment 4: In addition, please make your conclusions more generalised so that they can also be understood by a reader with no specialist knowledge of sport or physical culture.

Response to comment 4: Done as suggested.

Our results imply that biological age (i.e., maturity status) and not chronological age should be used to design training programs. Female adolescent badminton players should receive specifically targeted exercise interventions based on their fitness status and needs.

Comment 5: Supplementing the article with the above-mentioned scope will in my opinion make a real chance for publication in Sports. I keep my fingers crossed for the final success of the publication. 

Response to comment 5: Thank you very much for your helpful comments which definitely improved the quality of this paper.
